# Interleukin-6 in Rheumatoid Arthritis

**DOI:** 10.3390/ijms21155238

**Published:** 2020-07-23

**Authors:** Franco Pandolfi, Laura Franza, Valentina Carusi, Simona Altamura, Gloria Andriollo, Eleonora Nucera

**Affiliations:** 1Allergy and Clinical Immunology, Fondazione Policlinico Universitario A, Gemelli IRCCS, Catholic University, 00168 Rome, Italy; valentina.carusi@libero.it (V.C.); simona.altamura@guest.policlinicogemelli.it (S.A.); gloria.andriollo@libero.it (G.A.); eleonora.nucera@unicatt.it (E.N.); 2Emergency Medicine, Fondazione Policlinico Universitario A, Gemelli IRCCS, Università Cattolica del Sacro Cuore, 00168 Rome, Italy; cliodnaghfranza@gmail.com

**Keywords:** IL-6, rheumatoid arthritis, inflammation

## Abstract

The role of interleukin (IL)-6 in health and disease has been under a lot of scrutiny in recent years, particularly during the recent COVID-19 pandemic. The inflammatory pathways in which IL-6 is involved are also partly responsible of the development and progression of rheumatoid arthritis (RA), opening interesting perspectives in terms of therapy. Anti-IL-6 drugs are being used with variable degrees of success in other diseases and are being tested in RA. Results have been encouraging, particularly when anti-IL-6 has been used with other drugs, such as metothrexate (MTX). In this review we discuss the main immunologic aspects that make anti-IL-6 a good candidate in RA, but despite the main therapeutic options available to target IL-6, no gold standard treatment has been established so far.

## 1. Interleukin-6 in Health and Disease

Interleukin (IL)-6 is a prototypical cytokine, featuring pleiotropic and redundant functional activity. IL-6 belongs to a family of cytokines, which includes IL-6, IL-11, IL-27, IL-31, oncostatin M (OSM), leukaemia inhibitory factor (LIF), ciliary neurotrophic factor (CNTF), cardiotrophin 1 (CT-1), and cardiotrophin-like cytokine factor 1 (CLCF1); all these cytokines use the common IL-6 signal transducer gp130 [1]. IL-6 production can be induced by both infection and other types of inflammation. Indeed, IL-6 is promptly produced, mainly by macrophages, in response to pathogens or inflammation-related damage-associated molecular patterns [2] and performs a protective function by removing infectious agents and healing damaged tissue via induction of acute phase and immune responses. IL-6 is crucial for both innate and adaptive immunity. Several cell types produce IL-6, including monocytes, T-lymphocytes, fibroblasts, and endothelial cells, and production is strongly enhanced at sites of inflammation [3].

In infections, toll-like receptors (TLRs) directly recognize the bacteria, virus, or fungi and can directly or indirectly, depending on the type of stimuli, induce the production of IL-6 and other inflammatory cytokines, such as IL-1 or tumor necrosis factor α (TNFα), through the nuclear factor-kappa B (NF-kB) signaling pathway. Interestingly, the production of IL-1 and TNFα also stimulate the production of IL-6 [4]. Overall, dysregulated IL-6 production leads to persistent inflammation [5].

IL-6 demonstrates its biological activities only by binding to its specific receptor, IL-6R. Neither IL-6 nor IL-6R have affinity for gp130 (also known as CD130). However, as a complex, IL-6 and IL-6R can bind to and activate the IL-6Rβ-subunit, gp130, leading to its dimerization and intracellular signaling through Janus kinase (JAKs: JAK1, JAK2, and TYK2), in particular, signal transducer and activator of transcription (STAT)1 and 3. STAT1 and 3 are phosphorylated and translocated to the nucleus and induce the transcription of target genes in the nucleus, while SH2 domain-containing protein-tyrosine phosphatase 2 (SHP-2) activates the Ras-MAP kinase pathway [6]. While these molecules activate the transcription process, they also start a negative feedback signaling pathway, which suppresses cytokine signaling 1 (SOCS1) and SOCS3 proteins [4].

IL-6 was thought in the past to target only the liver, since hepatocytes express high IL-6R and gp130, but in recent years it has appeared more and more clear that IL-6 does not work on a single target organ [7]. IL-6 acts as an acute phase protein, enhancing the inflammatory response of the body. Most of its effects are indeed obtained through the liver, where it is processed and starts an inflammatory cascade, inducing C-reactive protein (CRP), serum amyloid A (SAA), fibrinogen, haptoglobin, and α1-antichymotrypsin. IL-6 also triggers a reduction in the levels of albumin, zinc, and iron through various mechanisms [5].

IL-6 also guides cell differentiation in the bone marrow, stimulating the production of neutrophils and megakariocytes, triggering liver production of thrombopoietin, which stimulates the differentiation of megakaryocytes in platelets, with IL-3 [8]. Indeed, like many other inflammatory agents, IL-6 production is also associated to anemia; its effects on hepatocytes include in-vitro production of hepcidin, not yet confirmed in vivo [9].

The effect of IL-6 on the immune system is particularly interesting; its effects are evident both on innate and acquired immunity. In terms of innate immunity, IL-6 is responsible for the maturation of inflammatory infiltrate, promoting neutrophil migration and mononuclear cell infiltration. It also acts as a chemoceptor for monocytes at the site of inflammation [10,11], while in terms of acquired immunity, IL-6 also expresses its action both on T-cells and on B-cells. On the latter, it has a differentiative effect towards active plasma-cells, increasing levels of serum gamma-globulins. Indeed, these two conditions are both present in Castelman’s disease, which is responsive to anti-IL-6 treatment [12]. During the 1980s, IL-6 was indeed known as B cell stimulating factor 2, yet, the role of IL-6 in the production of gamma-globulins, even though relevant, is not essential, as it is for other inflammatory mediators. Gamma-globulin production is indeed a complex process, in which different inflammatory mediators have redundant functions: IL-2 in particular is associated to immunoglobulin production, while IL-4 and TGF-β work as inhibitory molecules [13]. Interestingly, in other contexts, the effect of IL-6 and TGF-β are synergic.

The effects IL-6 has on T-cells are also quite relevant and associated to various pathologies. IL-6 acts mainly on the differentiation of CD4+ T-cells; in particular, it stimulates the Th17 pathway while inhibiting regulatory-T-cells (T-reg) differentiation. Th17 differentiation is important in fighting inflammation and promotes IL-6 production, further enhancing Th17 differentiation. Cytokines produced by Th17 lymphocytes also include TNFα, IL-1β, IL-17, IL-21, and IL-22, which stimulate inflammation and the fibrotic reaction of the tissue [14]. IL-6 on the other hand inhibits the production of T-reg cells through transforming growth factor β (TGFβ). In the right conditions, IL-6 is indeed an autoinflammatory mediator; not only does it reduce the capacity of the body to distinguish self from not self, it also promotes fibrosis and the inflammatory reaction. The importance of its role in many autoimmune diseases has been demonstrated and targeted [15,16].

## 2. Rheumatoid Arthritis and Immunologic Pathways

Rheumatoid arthritis (RA) is an autoimmune disease characterized by chronic inflammation affecting the joints and cartilage, which can lead to various degrees of osteoarthritis, and it can cause various degrees of disability. Even though the development and progression of RA is still not completely understood, different therapeutic options are available, and they have completely changed the prognosis of the disease [17]. Disease modifying antirheumatic drugs (DMARD), such as methotrexate (MTX), allow patients to keep the disease under control and lead normal lives. Unfortunately, not all patients respond to these drugs and their quality of life is severely impaired [18]. Factors involved in the development of RA can be roughly distinguished into environmental and genetic, with the former influencing the second

Overall, it is widely accepted that RA occurs in genetically predisposed individuals following the interaction with different environmental factors [19,20]. The presence of T-cell receptor (TCR) restriction at the joints has also been described, suggesting T-cell attack of a joint antigen. It may also prove important in terms of predicting poor response to conventional therapy and severity of disease [21,22].

The common amino acid motif (QKRAA) in the HLA-DRB1 region, commonly known as the shared epitope, confers particular susceptibility [23]. Given the importance in determining T-cell repertoire of this region, the central role of T-cells appears obvious.

Joint damage results from the degradation of connective tissue by tissue-destroying matrix metalloproteinases (MMP) and the stimulation of osteoclastogenesis through the receptor activator of the nuclear factor-kB ligand (RANKL).

Among the environmental factors for RA, smoking is an important one, causing alterations in the oral microbiota, which might trigger autoimmune cross-reactions [24]. Something similar also happens with some infections, such as Epstein-Barr and cytomegalovirus infections: These infections over-stimulate the immune system, favoring suppression of T-regs and proliferation of Th17 lymphocytes [25]. In susceptible individuals, the chronic exposure to citrullinated peptides in conditions of chronic inflammation, as in the above-mentioned infections, is a key step in the pathogenesis of RA [26]. The role of the microbiota also needs to be acknowledged: The microbiota is a well-known regulation agent of inflammation and immunity, as it can act as a pro-inflammatory stimulus or reduce systemic inflammation, also modifying the pattern of cytokine production of the organism [27]. In RA, oral microbiota appears to be particularly significant. Indeed, it stimulates both production of aberrant neutrophils, which appear to play, in general, a central role in RA, and also favors cross-reactivity with proteins expressed in the joints [28]. *Lactobacillus salivarius* is the most represented bacteria in patients suffering from RA, and molecular mimicry of antigens involved in RA has been described. Interestingly, once patients are treated for RA, their oral microbiota changes and becomes similar to that of the general population [29].

From an immunologic point of view, both the adaptive and the innate immune systems are involved. Neutrophils and monocytes are the most important actors as far as innate immunity is concerned in the pathogenesis of RA, as they directly influence the interaction between toll-like-receptors (TLR) and other signaling molecules. In particular, myeloid-related protein (MRP) 8/14 further activates monocytes and creates an inflammatory environment [30]. The role of macrophages has also been acknowledged: They are capable of stimulating TNF and IL-1β production and they activate a wide number of immune effectors. They are also important in terms of clinical presentation, as some of the enzymes they produce are directly involved in cartilage deterioration [31]. As for the adaptive immunity, both B-lymphocytes and T-lymphocytes are involved.

The importance of B-cells in RA has been questioned in the past, but has been confirmed by the efficacy of treating affected patients with rituximab, an anti-CD20 antibody [32]. Plasma cells, which produce antibodies, do not express CD20, proving that the role of B-cells goes beyond autoimmune antibody production. Interestingly, activated CD4+ T cells also stimulate B cells to produce immunoglobulins.

### TH17 in RA

The Th1 pathway is traditionally considered the most important in RA, while Th17 CD4 lymphocytes have only recently been acknowledged as important mediators in the pathogenesis of RA [14]. Precursor Th17 lineage cells expressing CD161 or lectin-like transcript 1 (LLT1) accumulate in patients suffering from RA, playing a central role in the pathogenesis [33]. IL-17 production is central in stimulating synovial fibroblasts and it also stimulates osteoclastogenesis in a RANKL-dependent and RANKL-independent fashion: In the first pathway, it acts directly to produce osteolysis. IL-17 also induces osteoblasts to produce RANKL, which act in synergy with TNF-α, resulting in increased osteoclastogenesis. Th-17 also produces IL-21 and 22 and TNF-α, in addition to IL-17, potentially inducing a vicious circle of inflammation [34]. The activation of all these mechanisms seems to be determined at least in part by IL-6, which also determines a reduction in the number of T-regs. Interestingly, IL-6 is further stimulated by IL-17, further maintaining inflammation. The imbalance between the inflammatory pathway and the suppressive one is, indeed, central in the pathogenesis of RA: T-regs in patients with RA appear not only reduced in quantity, but also in quality [35].

In RA, the signaling pathway of the TLRs and the expression of inflammatory cytokines (e.g., IL-1, TNFα, and IL-17) work as IL-6 promoter-activators, as do other factors [36]. Once at the joint, IL-6 has a crucial role in the inflammatory process, in osteoclast-mediated bone resorption and in pannus development; these processes help the development of IgM and IgG rheumatoid factors along with antibodies to citrullinated peptides, which are characteristically increased in RA [37]. At a joint level, IL-6 works on many different paths; its importance has been evidenced in obese patients suffering from osteoarthritis. Obesity is linked to high levels of inflammatory proteins, such as IL-6, in the blood, but also, for instance in the synovial liquid. At this level, inflammatory proteins favor the production of MMPs, particularly, MMP-1, MMP-3, MMP-13, and aggrecanase 1 and 2 (ADAMTS-4, ADAMTS-5) [38]. As previously discussed, MMP activation is also key in the progression of RA.

Other immunologic pathways are also involved in the development of RA and can eventually lead to IL-6 overexpression. For instance, the nuclear factor kappa-light-chain-enhancer of activated B cells (NF-kB) pathway is a key inflammatory mediator in RA, determining an increase in TNFα, which in turn increases IL-6 levels.

IL-6 might also be involved in a more subtle way in the development of RA: It has been observed that IL-6 is involved in cytokine release syndrome complicated by T-cell therapy. Blocking IL-6 in these cases gave good results, proving the central part it plays in inflammatory syndromes [39,40]. IL-6 seems to be responsible for other systemic symptoms associated with RA, in particular in the nervous and cardiovascular systems.

Overall, in patients with RA, elevated serum levels of both IL-6 and IL-6R are found in serum and synovial fluid of affected joints [41].

Other important cytokines that are currently being studied as possible therapeutic targets are IL-17 and granulocyte-macrophage colony-stimulating factor (GM-CSF) [42], further demonstrating the interaction between both innate and adaptive immunity.

Even though RA is usually thought of as a disease affecting mostly the joints, patients who suffer from this disease often face a number of comorbidities and are at a higher risk of developing certain types of disease: Cardiovascular disease is particularly common in this population, for instance, and this seems to be directly affected by IL-6 [43]. A similar situation has been described for psychiatric disorders. Patients suffering from chronic conditions, autoimmune diseases in particular, are known to be at a higher risk of developing a broad range of psychiatric disorders, particularly depression. In this group of patients, those with higher levels of IL-6 and CRP seem to be at a higher risk of developing psychiatric comorbidities [44]. Thus, targeting IL-6 might improve overall health in patients suffering from RA.

Overall, inflammatory cytokines, especially TNF-α, and two interleukins, IL-1β and IL-6, are key in driving inflammation and joint damage. Cytokines such as IL-23, IL-17A, as discussed above, and interferon gamma (IFN-γ) also play crucial roles in the pathogenesis of RA. IL-4 and IL-10, on the other hand, have been suggested to improve arthritis [45]. While there is overall consensus on the inflammatory role of IL-6 in the pathogenesis of RA, there are some studies pointing out that its role is not clear-cut: It has emerged that IL-1 signaling reduces IL-6 signaling in RA, overall worsening patients’ conditions. The possibility of acting directly on IL-1 is thus interesting; at the moment, the most interesting results have been found for patients suffering from other chronic inflammatory diseases, such as type-2 diabetes, in which patients whose IL-1 pathway is blocked may have benefits both in RA and diabetes control [46]. IL-6 acts differently on different cell types, and it can even inhibit fibrosis, but when in a crosstalk with IL-1, its inflammatory characteristics seem to overtake its anti-inflammatory role [47]. Yet, this hypothesis is based primarily on in vitro studies and needs to be confirmed further.

## 3. Interleukin-6 in Rheumatoid Arthritis and Other Diseases

The IL-6 pathway is involved in different inflammatory diseases and could be a potential target in a vast array of clinical conditions in different branches of medicine [48] spanning RA, cytokine released syndrome in CART cells [49], or COVID-19 infection [50,51].

IL-6 blockade has been successfully used in Castelman’s disease, a lymphoproliferative disorder with a wide spectrum of manifestations, and juvenile idiopathic arthritis (JIA). In some countries it has also been used with various degrees of success in other rheumatologic diseases, such as giant cell arteritis (GCA), Takayasu arteritis (TA), and cytokine releasing syndrome (CRS).

In the past few months, IL-6 has gained visibility because of its effects on COVID-19 infection. COVID-19 is a disease caused by a novel coronavirus and it can cause a vast array of symptoms, ranging from mild, flu-like symptoms to severe respiratory failure [52]. Many different therapeutic approaches have been tried, mostly anti-inflammatory therapies, ranging from steroids to antimalaria drugs; even non-conventional therapies, such as ozone-therapy, have been tested [53]. The most severe symptoms seem to be caused, at least in part, by an autoinflammatory reaction, with characteristics of a cytokine storm [52]: Binding of the novel coronavirus to TLRs causes an increase in the levels of IL-1β, which mediate fibrosis and inflammation of the lung [50]. The role of IL-6 in promoting fibrosis has been studied in other pathologies, particularly in the liver [54]. Blocking IL-6 in this context has proven useful: Patients have been treated with tocilizumab (TCZ), a monoclonal antibody against IL-6R, and results have been encouraging [51]. The role played by IL-6 in COVID-19-related inflammation is confirmed by the fact that patients with higher circulating levels of IL-6 and other inflammatory cytokines, such as persons suffering from Down syndrome, are at a higher risk of developing more severe forms of COVID-19 infection [55].

In the treatment of RA, IL-6 blockade has proven to be very useful for those patients who do not respond to conventional therapy or even less standard ones. For instance, TNF-α had been a promising target in patients with severe RA, but up to two thirds of patients do not have a full response; in this group, targeting IL-6 is not only useful, but even improves the overall response to therapy of these patients [56].

An increase in serum IL-6 and IL-21 levels is associated with markers of B cell activation, and IL-6 is associated with radiographic progression in patients with RA [57]. Based on studies emphasizing the critical role of IL-6 in development and progression of RA, targeting IL-6 has been proposed as a tool to add to the armamentarium to treat RA. Targeting IL-6 can be achieved by either direct targeting of the cytokine or targeting of its receptor. Targeting the receptor (instead of the cytokine itself) may have the advantage of blocking other cytokines of the IL-6 family.

## 4. Targeting Interleukin-6: Available Options

As discussed above, targeting IL-6 has proven effective in many contexts. Drugs targeting IL-6 can be roughly divided into direct inhibitors and IL-6 receptor inhibitors. The latter offer an advantage, as they have the capacity to also block other interleukins of the same family, such as IL-1, which also play an important role in the pathogenesis of RA [58].

Different types of IL-6 inhibitors will be discussed in the following paragraphs.

TCZ was introduced in 2008 and reduced disease activity in RA with inadequate response to DMARD [59].

In the last few years, several trials in over 23,700 patients have confirmed the efficacy and safety of TCZ (alone or in combination with DMARD) in RA [36], demonstrating the higher efficacy of a combination of DMARDs than MTX on its own.

The randomized OPTION study on 622 patients proved that the use of TCZ reduces signs and symptoms of RA, proving that the inhibition of IL-6 signaling is effective in moderate-to-severe RA [60].

In the TAMARA trial, 286 patients with moderate to severe RA were treated with TCZ and it was concluded that TCZ is effective in improving patients’ condition, the most solid results being seen at about 24 weeks [61].

Different routes of administration of TCZ have also been studied, demonstrating that the sub-cutaneous (SC) route is also safe [59,62,63,64,65,66,67]. Extensive experience has firmly established the short- and long-term efficacy of both intravenous and SC TCZ in adults with moderate-to-severe RA who failed to respond to therapy with synthetic or biologic DMARDs (review in [62]). In one of the first pivotal studies (SAMURAI), it was shown that in 306 patients with active RA, TCZ monotherapy was generally well tolerated and provided radiographic benefit [68]. Another important study, AMBITION, randomized 673 patients, showing that TCZ was better than MTX [69].

The large LITHE study further demonstrated the disease-modifying effects on IL-6R antagonism. The study enrolled 1196 patients, suffering from RA, ranging from moderate to severe forms of the disease. All participants did not show an adequate response to methotrexate (MTX) therapy. During the study, all patients were treated with MTX but they were also randomly divided in a placebo group and a group receiving TCZ (4 or 8 mg/kg, IV) every 4 weeks in combination with MTX. At week 52, patients who were administered TCZ 8 mg/kg showed a mean change in the Genant-modified Sharp, demonstrating a significantly lower radiographic progression of the disease when compared to those on the placebo group (0.29 vs. 1.13; *p* < 0.0001) [70], further confirming the safety and effectiveness of TCZ. The incidence of adverse effects was 1.2% in patients with no risk factors and 11.2% in patients with three or more risk factors. The most frequent adverse effects were infections, particularly respiratory ones, both bacterial and viral [71]. In this regard, a separate study, which enrolled 63 RA patients, showed that a short course of TCZ increased the risk of hepatitis B reactivation [63].

Another recent paper by Rutherford, 2018 [72] compared the incidence of serious infections in RA patients treated with different DMARDs, even though there was little directly comparative literature of infection risk between TCZ and other drugs. Kerschbaumer et al. studied 19,282 patients with 46,771 global years of follow-up and reported that the incidence of serious infections (SI) was 5.51 cases per 100 patient years for the entire cohort (95% confidence interval (CI) 5.29 to 5.71). Compared with etanercept, TCZ had a higher risk of serious infections (HR 1.22, 95% CI 1.02 to 1.47) especially sepsis and respiratory infections ([73]).

Overall, TCZ has a well-characterized and tolerable safety profile [70].

Another IL-6 R inhibitor, sarilumab (SAR), has also been approved in RA in combination with MTX [74,75,76,77,78]. In the MONARCH study, it was demonstrated that SAR monotherapy was more efficient than adalimumab (an anti-TNFα) monotherapy, improving signs and symptoms and physical functions in patients with RA who could not continue taking MTX [79]. In addition, atlizumab, an IL-6R inhibitor, was proven to be beneficial [80].

As stated above, targeting IL-6 can been achieved by biological reagents, namely monoclonal antibodies directed against IL-6 itself such as siltuximab [81], which is approved for multicentric Castleman’s disease [82] and has shown efficacy in RA [83,84].

Sirukumab is another anti-IL-6 drug, and it has been approved for Castelman’s disease, with positive preliminary data in RA. The SIRROUND-D study showed improvements both in the Clinical Disease Activity Index and clinically. Meaningful improvements in patient-reported outcomes were sustained at week 104 among patients initially randomized to sirukumab [85]. Another randomized study in 38 healthy patients showed similar results, confirming sirukumab’s well-tolerated safety profile [86].

However, the license request for treatment in RA was denied due to excess mortality in the sirukumab arm in comparison to placebo: On 2 Auguest 2017 the Food and Drug Administration (FDA Advisory Committee Meeting denied approval. The majority of the committee (11 versus 2) agreed that it was not clear whether the imbalance in all-cause mortality is actually a side-effect of the drug or rather a consequence of the design of the study. Overall, the committee agreed that more studies should be conducted with sirukumab because, the “safety profile seemed to be consistent with a class effect showing similar adverse effects and laboratory abnormality profiles. However, death rates in sirukumab arms compared with placebo, especially in the controlled period, raised safety concerns, which led to the decision by the FDA to decline the approval of SRK in August 2017. The majority of the committee (11 to 2) did not agree that the safety profile of SRK 50 mg SC every 4 weeks is adequate to support the approval of sirukumab for the treatment of adult patients with moderately to severely active RA who have had an inadequate response or are intolerant to one or more DMARDs”.

Following this decision, the company, albeit convinced that more data may show its efficacy, stopped pursuing this line of research [87].

Clazakizumab is also an anti-IL-6 drug with promising results. In a study with 418 participants, patients with RA received SC clazakizumab once a month at 25, 100, or 200 mg plus MTX, once-monthly only SC clazakizumab at 100 or 200 mg as monotherapy, or MTX plus a placebo (in other words, MTX alone). Patients who received clazakizumab had a significantly greater response at week 12 when compared to patients receiving MTX alone [88].

Olokizumab, also an anti-IL-6 reagent, gave better results than placebo in 119 randomized patients [89].

In addition to the above-mentioned agents, a number of reagents against IL-6R and IL-6 have been developed and used in RA (Table 1).

An IL-6 blockade has proven to be effective not only in classic RA, but also in adult onset Still’s disease. Still’s disease is a rare generalized form of RA, which can present with a vast array of symptoms. Given its rarity, no systemic studies have been conducted on therapeutic strategies, and therapies are based mostly on clinicians’ experience. Targeting IL-1 and TNF-α has proven effective in the context of the articular symptoms of the disease, but, to date, only IL-6 has proven consistently effective in treating systemic symptoms [90].

The safety and efficacy of an IL-6 blockade has also been demonstrated in patients suffering from JIA, a form of RA affecting children and young adults. There are many forms of JIA, some of which closely resemble adult RA, but therapeutic strategies are limited by the young age of patients and differences in the immune systems of children and adults. DMARDs normally used in RA do not always offer satisfactory results. Recently, though, TCZ has been tested in this population and results have been encouraging, demonstrating the central role of IL-6 in autoimmune diseases [91].

Overall, both direct and indirect IL-6 inhibitors have demonstrated acceptable safety profiles, even though some studies have pointed out that mortality rates associated to direct IL-6 inhibition are higher than the ones associated to those of an IL-6 receptor blockade. Studies highlighting this discrepancy, though, compared the mortality rates between those taking an IL-6 inhibitor to those under placebo, so the interpretation of these data is not clear [58].

## 5. Conclusions

The development of a vast array of reagents against IL-6 and IL-6R is a demonstration per se that the perfect drug to treat RA has not been found yet. Interesting results have indeed been obtained by treating patients with different IL-6 antagonists in combination with different DMARDs: Patients showed improvement in terms of symptoms and quality of life, further proving the central role of IL-6 in the pathogenesis of RA.

Targeting IL-6 or its receptor is a safe and effective treatment, but further studies are required to identify the gold standard of treatment, because direct drug-to-drug comparison trials are still lacking and, even though scarce, there is information suggesting that the safety profile of these drugs may not be fitting for all patients and may even vary from drug to drug. Overall, in those patients who do not adequately respond to classic therapy, anti-IL-6 targeting is an interesting strategy that needs to be evaluated, given its potential capacity to completely modify patients’ prognosis and reduce the burden of disease.

## Figures and Tables

**Table 1 ijms-21-05238-t001:** IL-6 and IL-6R mAbs.

Biological	Other Names	Commercial Name	Producer	Target
Siltuximab		Sylvant **		IL-6
Clazakizumab	LD518 and BMS-945429		Adler Biopharma	IL-6
Olokizumab			UCB	IL-6
Vobarilizumab				Il-6R
Olamkicept	FE-301; FE-999301; TJ-301		Ferring	IL-6R
Sarilumab		Kevzara	Sanofi	IL-6R
Sirukumab **			Janssen	IL-6
Olamkicept			Ferring	
Satralizumab	Sapelizumab, SA237		Roche	IL-6R
Tocilizumab		Tocilizumab, Actemra e RoActemra.	Roche	IL-6R

** this drug is approved for multicentric Castleman disease.

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
