# Peer review of "Interleukin-6 in Rheumatoid Arthritis"

_ijms, 2020, doi:10.3390/ijms21155238_

Round 1

Reviewer 1 Report

The manuscript Interleukin-6 in Rheumatoid Arthritis by Pandolfi et al. is a thorough, updated and exhaustive review of the literature on the role of IL6 and its therapeutic inhibition in the pathogenesis of rheumatoid arthritis. The manuscript is well written, thus the intervention of this reviewer will only be limited to some spelling and formal remarks.

line 26: “cytokine” should be “cytokines”;

lines 59-60: “IL-6 is responsible for the maturation of neutrophils towards mononuclear cells” should be “IL-6 amplify the leucocyte recruitment”

line 84: “disease[15].” should be “disease [15].”;

lines 84, 218, 220, 221, 245, 309: “DMART” should be “DMARD”;

line 95: “point view” should be “point of view”;

line 99: “whit” should be “with”;

line 125: “acumulate” should be “accumulate”;

line 161: “offer” should be “often” and “a t” should be “at”;

line 187: “is” should be “has”;

line 196: “lung[46]” should be “lung [46]”;

line 200: “ahigher” should be “a higher”;

line 230: the whole line should be cancelled

line 242: “particularly respiratory ones, but also viral” should be “both, bacterial and viral”

line 247: “thatthe” should be “that the”;

line 262: “Another a” should be “Another”;

line 280: “MTXonce” should be “MTX once”;

line 288: "this drug is approved for approved for multicentric Castleman disease" should be "this drug is approved for multicentric 288 Castleman disease"

References 27, 49, 67, 68, 69, 75, 81, and 87 are incomplete, thus they should be completed.

Author Response

Thank you for your suffestion. We have addressed each comment in detail below.

line 26: “cytokine” should be “cytokines”;

We modified it

lines 59-60: “IL-6 is responsible for the maturation of neutrophils towards mononuclear cells” should be “IL-6 amplify the leucocyte recruitment”

We modified the sentence

line 84: “disease[15].” should be “disease [15].”;

We corrected the mistake

lines 84, 218, 220, 221, 245, 309: “DMART” should be “DMARD”;

We corrected the mistake

line 95: “point view” should be “point of view”;

We corrected the mistake

line 99: “whit” should be “with”;

We corrected the mistake

line 125: “acumulate” should be “accumulate”;

We corrected the mistake

line 161: “offer” should be “often” and “a t” should be “at”;

We corrected the mistake

line 187: “is” should be “has”;

We corrected the mistake

line 196: “lung[46]” should be “lung [46]”;

We corrected the mistake

line 200: “ahigher” should be “a higher”;

We corrected the mistake

line 230: the whole line should be cancelled

We corrected the mistake

line 242: “particularly respiratory ones, but also viral” should be “both, bacterial and viral”

We corrected the mistake

line 247: “thatthe” should be “that the”;

We corrected the mistake

line 262: “Another a” should be “Another”;

We corrected the mistake

line 280: “MTXonce” should be “MTX once”;

We corrected the mistake

line 288: "this drug is approved for approved for multicentric Castleman disease" should be "this drug is approved for multicentric 288 Castleman disease"

We corrected the mistake

References 27, 49, 67, 68, 69, 75, 81, and 87 are incomplete, thus they should be completed.

We corrected the mistake (49 is a preprint and no journal is available)

Reviewer 2 Report

* General recommendation: There is a need for improvement to the English language within your manuscript.

Line 15_ In the abstract "Castelman's disease" is specifically cited but throughout the review it is not described or explained what the disease consists and neither is it a topic that is developed further. From my point of view, this disease should not be included in the abstract.

Lines 36-38_ The information described in these lines has already been introduced in lines 27-29. It would be useful to incorporate this small paragraph after lines 27-29.

Lines 39-45_It's too long a sentence. It is advisable to divide it into two shorter sentences.

Lines 45-46_ The idea expressed in the sentence “ …, they also start a negative feedback signaling pathway, which are suppressor of cytokine signaling 1 …”  is not understood. There is no concordance between "a negative feedback" and "which are suppressor" (?).

Linea 51_Maybe instead of "including" you should have written "inducing".

Lines 51-53_ The effects of IL-6 on mediators are described in these lines, but it would be interesting to summarize the physiological consequences of these changes.

Line 55_”Thrombopoietin” must be written in lowercase

Paragraph lines 54-57_ Reference 9 does not include such information with respect to thrombopoietin. In Ref.9,  this mediator is only mentioned as part of a hypothesis proposed by other research groups  [ “ Data from several laboratories have suggested that there are growth factors (designated "potentiators", thrombopoiesis-stimulatingfactorand thrombopoietin) that synergize with defined colony-stimulating factors to promote megakaryocytic colony formation (although these factors do not primarily induce colony formation), and/or promote megakaryocytic maturation (33-41). Because these factors have not been purified, their relationship to IL-6 is unknown.”]

In this same paper (Ref.9)  it is pointed out that:  “Although IL-6 promotes maturation of megakaryocytes in vitro, its influence on megakaryocyte growth and, ultimately, platelet production in vivo is unknown.” This limitation of the conclusions to in vitro experiments should be noted in the text of the review.

Line 60_ The phrase is ambiguous, suggesting that neutrophils are cells that mature into mononuclear cells. The wording is confusing. IL-6 is actually involved in the evolution of the type of inflammatory infiltrate during the course of inflammation. IL-6 orchestrates transition from the neutrophil to mononuclear-cell infiltrate by enhancing neutrophil migration.

Line 63_ Basic information on Castelman's disease should be included, since it is cited throughout the work without the basic characteristics of the pathology having been described.

Line 72_ The term regulatory T cells and its abbreviation have not previously appeared. It would be useful to write regulatory T cells and its abbreviation in brackets. Besides, the full term "regulatory T cells" appears below in line 75. A new term "regulatory T-cells" appears in line 101. In line 134 appears as "T-regs". Please correct and unify all terms.

Line 73_ The expression “works as a positive feedback” is hard to understand. When "positive feedback" is used, it is usually referred to “feedback loop” or “feedback system”.

Lines 75-76_The reference from which the information has been extracted has not been indicated.

Lines 78-79… It would be interesting to expand on this idea, otherwise it is very general information and does not provide any relevant data (specify the works from which the information was extracted).

Line 81 _ “affecting the joints and cartilage” is a vague description of RA as it is a characteristic more commonly associated with osteoarthritis.

Lines 81-82_ Line 81 should indicate that the abbreviation for rheumatoid arthritis is RA. The abbreviation has not yet appeared in the introdcution section.

Line 84_ The abbreviation used for Disease-modifying anti-rheumatic drugs should be DMARD and not DMART. Please correct de abbreviation in lines 84, 218, 220, 221, 245,309.

Lines 88-89_ These lines do not express the ideas reflected in the conclusions of the works cited in references 19 and 20. This information should be reviewed and corrected.

Lines 93-94_ This text is a continuation of the idea reflected in lines 87-88, so it should be integrated into the previous paragraph.

Lines 95-97_ The idea expressed in these lines is related to the content of reference 19. The organization and content of the 3 paragraphs between lines 87-97 should be reformulated.

Lines 113-116_ It is an excessively long sentence, it is difficult to understand the idea expressed. Furthermore, when discussing innate immunity, the role of macrophages in RA should be mentioned.

Lines 116-121_ Information on the role of innate immunity is very scarce.

Lines 122_ Maybe instead of "TH17" you should have written "Th17".

Lines 126-128_ It is recommended to simplify the writing of the statement regarding the effects of IL-17 on fibroblasts and on osteoclastogenesis. It may be helpful to separate the ideas into 2 sentences.

Line 132_ Please correct the IL-17 dash.

Line 138_ The term “ bone re-absorption” is correct but "bone resorption" is used preferably.

Lines 145,146,148… It is a bit confusing to refer to rheumatoid arthritis in the text using its abbreviation (RA) or its full script without any apparent criteria. Please unify the writing.

Line 152_ Please rephrase the wording of the sentence “…proving the central part IL-6 plays in many other inflammatory …” as it is not understood. There seems to be some missing words.

Lines 155-156_ Bibliographic reference should be indicated.

Line 168_ The abbreviation of TNF alpha has already been noted on line 33.

Line 169_ Maybe instead of " IL-1B " you should have written " IL-1 beta".

Lines 179-181_This paragraph is a continuation of the last idea raised in the previous paragraph. It should not be separated.

Line 177, line 181_ standardize writing of “in vitro” and “in-vitro”

Line 185_ The term “Car-T cells” is usually spelled in capital letters (“CAR T cells”)

Lines 194-196_ Please review the information regarding the synthesis of IL6 included in the following sentence: “…binding of the novel coronavirus to TLRs causes an increase in the levels of IL-1β and -6, which in turn favors fibrosis and inflammation of the lung [46].”

The paper in ref. 46 specifies that IL-1 levels increase after TLR activation, but does NOT mention IL6 synthesis [ “ The binding of COVI-19 to the Toll Like Receptor (TLR) causes the release of pro-IL-1β which is cleaved by caspase-1, followed by inflammasome activation and production of active mature IL-1β which is a mediator of lung inflammation, fever and fibrosis.”]

Line 201_ There is a missing word in the following sentence: “..IL-6 blockade has proven very useful for those “ (“has been proven very useful”)

Lines 182 and 211_ Unify the writing of IL-6 as Interleukin-6 or IL-6 in the headings of the text sections.

Linea 197_ The term “Tocilizumab” appears for the first time in the text, include its abbreviation here instead of in line 217.

Linea 217_ The description of “Tocilizumab” had already been detailed in line 197. This is redundant information.

Line 229_ Please specify in the text the meaning of the abbreviations "IV" and "SC" (intravenous (IV) and subcutaneous (SC)). In lines 279-280 the abbreviation for subcutaneous (SC) is indicated twice.

Line 236_ The abbreviation for methotrexate (MTX) already indicated on line 84.

Lines 237-239_ These lines contain details of the commented experiments that are not relevant and make the text unnecessarily dense. Only the conclusion of the cited study should be included.

Lines 251-252_Why Etanercept, Tocilizumab in uppercase and sarilumab and adalimab in lowercase?. In the paragraph 260-263, Sirukumab is written in both upper and lower case (??). Please, unify the writing.

Line 254_“The therapeutic target of the atlizumab is not explained. Please clarify it.

Line 261_ Why is "Week 104" written in uppercase?

Line 270_The FDA text cited refers to sirukumab with its abbreviation SRK, but it has not previously appeared in the review. It would be recommendable to indicate the abbreviation previously in the text to improve the understanding.

Author Response

Thank you for your comments and suggestions. We will further address the issues you underlined below.

Line 15_ In the abstract "Castelman's disease" is specifically cited but throughout the review it is not described or explained what the disease consists and neither is it a topic that is developed further. From my point of view, this disease should not be included in the abstract.

We have described Castelman’s disease in the paper, as it is the main therapeutical indication of anti-IL6 drugs. Nevertheless, we have eliminated it from the abstract

Lines 36-38_ The information described in these lines has already been introduced in lines 27-29. It would be useful to incorporate this small paragraph after lines 27-29.

We have followed your suggestion

Lines 39-45_It's too long a sentence. It is advisable to divide it into two shorter sentences.

We have followed your advice

Lines 45-46_ The idea expressed in the sentence “ …, they also start a negative feedback signaling pathway, which are suppressor of cytokine signaling 1 …”  is not understood. There is no concordance between "a negative feedback" and "which are suppressor" (?).

We modified the sentence accordingly

Linea 51_Maybe instead of "including" you should have written "inducing".

We have modified it accordingly

Lines 51-53_ The effects of IL-6 on mediators are described in these lines, but it would be interesting to summarize the physiological consequences of these changes.

We have shortly described its effects

Line 55_”Thrombopoietin” must be written in lowercase

We have modified it

Paragraph lines 54-57_ Reference 9 does not include such information with respect to thrombopoietin. In Ref.9,  this mediator is only mentioned as part of a hypothesis proposed by other research groups  [ “ Data from several laboratories have suggested that there are growth factors (designated "potentiators", thrombopoiesis-stimulatingfactorand thrombopoietin) that synergize with defined colony-stimulating factors to promote megakaryocytic colony formation (although these factors do not primarily induce colony formation), and/or promote megakaryocytic maturation (33-41). Because these factors have not been purified, their relationship to IL-6 is unknown.”]; In this same paper (Ref.9)  it is pointed out that:  “Although IL-6 promotes maturation of megakaryocytes in vitro, its influence on megakaryocyte growth and, ultimately, platelet production in vivo is unknown.” This limitation of the conclusions to in vitro experiments should be noted in the text of the review.

We added a reference and added the limitation you pointed out to the text.

Line 60_ The phrase is ambiguous, suggesting that neutrophils are cells that mature into mononuclear cells. The wording is confusing. IL-6 is actually involved in the evolution of the type of inflammatory infiltrate during the course of inflammation. IL-6 orchestrates transition from the neutrophil to mononuclear-cell infiltrate by enhancing neutrophil migration.

We rewrote the sentence to make it clearer.

Line 63_ Basic information on Castelman's disease should be included, since it is cited throughout the work without the basic characteristics of the pathology having been described.

We have addressed the issue.

Line 72_ The term regulatory T cells and its abbreviation have not previously appeared. It would be useful to write regulatory T cells and its abbreviation in brackets. Besides, the full term "regulatory T cells" appears below in line 75. A new term "regulatory T-cells" appears in line 101. In line 134 appears as "T-regs". Please correct and unify all terms.

We have done as recommended.

Line 73_ The expression “works as a positive feedback” is hard to understand. When "positive feedback" is used, it is usually referred to “feedback loop” or “feedback system”.

We have reworded the sentence

Lines 75-76_The reference from which the information has been extracted has not been indicated.

We have added the reference

Lines 78-79… It would be interesting to expand on this idea, otherwise it is very general information and does not provide any relevant data (specify the works from which the information was extracted).

We have added some references to further explain the concept.

Line 81 _ “affecting the joints and cartilage” is a vague description of RA as it is a characteristic more commonly associated with osteoarthritis.

We described gave a more precise definition of the disease.

Lines 81-82_ Line 81 should indicate that the abbreviation for rheumatoid arthritis is RA. The abbreviation has not yet appeared in the introdcution section.

We have done as suggested.

Line 84_ The abbreviation used for Disease-modifying anti-rheumatic drugs should be DMARD and not DMART. Please correct de abbreviation in lines 84, 218, 220, 221, 245,309.

We have done as suggested.

Lines 88-89_ These lines do not express the ideas reflected in the conclusions of the works cited in references 19 and 20. This information should be reviewed and corrected.; Lines 93-94_ This text is a continuation of the idea reflected in lines 87-88, so it should be integrated into the previous paragraph.; Lines 95-97_ The idea expressed in these lines is related to the content of reference 19. The organization and content of the 3 paragraphs between lines 87-97 should be reformulated.

We have modified this part accordingly

Lines 113-116_ It is an excessively long sentence, it is difficult to understand the idea expressed. Furthermore, when discussing innate immunity, the role of macrophages in RA should be mentioned.; Lines 116-121_ Information on the role of innate immunity is very scarce.

We have changed the sentence and shorty described the role of macrophages.

Lines 122_ Maybe instead of "TH17" you should have written "Th17".

We have changed it accordingly

Lines 126-128_ It is recommended to simplify the writing of the statement regarding the effects of IL-17 on fibroblasts and on osteoclastogenesis. It may be helpful to separate the ideas into 2 sentences.

We have followed your advice

Line 132_ Please correct the IL-17 dash.; Line 138_ The term “ bone re-absorption” is correct but "bone resorption" is used preferably.; Lines 145,146,148… It is a bit confusing to refer to rheumatoid arthritis in the text using its abbreviation (RA) or its full script without any apparent criteria. Please unify the writing.

We have addressed these issues in the text

Line 152_ Please rephrase the wording of the sentence “…proving the central part IL-6 plays in many other inflammatory …” as it is not understood. There seems to be some missing words.

We modified the sentence

Lines 155-156_ Bibliographic reference should be indicated.

We have added one

Line 168_ The abbreviation of TNF alpha has already been noted on line 33.; Line 169_ Maybe instead of " IL-1B " you should have written " IL-1 beta". ; Lines 179-181_This paragraph is a continuation of the last idea raised in the previous paragraph. It should not be separated.; Line 177, line 181_ standardize writing of “in vitro” and “in-vitro”; Line 185_ The term “Car-T cells” is usually spelled in capital letters (“CAR T cells”)

We have addressed these issues

Lines 194-196_ Please review the information regarding the synthesis of IL6 included in the following sentence: “…binding of the novel coronavirus to TLRs causes an increase in the levels of IL-1β and -6, which in turn favors fibrosis and inflammation of the lung [46].” The paper in ref. 46 specifies that IL-1 levels increase after TLR activation, but does NOT mention IL6 synthesis [ “ The binding of COVI-19 to the Toll Like Receptor (TLR) causes the release of pro-IL-1β which is cleaved by caspase-1, followed by inflammasome activation and production of active mature IL-1β which is a mediator of lung inflammation, fever and fibrosis.”]

We have revised the sentence and added a reference

Line 201_ There is a missing word in the following sentence: “..IL-6 blockade has proven very useful for those “ (“has been proven very useful”)

We have modified the sentence

Lines 182 and 211_ Unify the writing of IL-6 as Interleukin-6 or IL-6 in the headings of the text sections.

We have unified the headings

Linea 197_ The term “Tocilizumab” appears for the first time in the text, include its abbreviation here instead of in line 217.; Linea 217_ The description of “Tocilizumab” had already been detailed in line 197. This is redundant information.

We have modified the text accordingly

Line 229_ Please specify in the text the meaning of the abbreviations "IV" and "SC" (intravenous (IV) and subcutaneous (SC)). In lines 279-280 the abbreviation for subcutaneous (SC) is indicated twice.

We have addressed the comment

Line 236_ The abbreviation for methotrexate (MTX) already indicated on line 84.

We corrected the mistake

Lines 237-239_ These lines contain details of the commented experiments that are not relevant and make the text unnecessarily dense. Only the conclusion of the cited study should be included.

We have modified these paragraphs

Lines 251-252_Why Etanercept, Tocilizumab in uppercase and sarilumab and adalimab in lowercase?. In the paragraph 260-263, Sirukumab is written in both upper and lower case (??). Please, unify the writing.

We have unified the writing

Line 254_“The therapeutic target of the atlizumab is not explained. Please clarify it.

We have added a short description

Line 261_ Why is "Week 104" written in uppercase?

It was a mistake and we corrected it

Line 270_The FDA text cited refers to sirukumab with its abbreviation SRK, but it has not previously appeared in the review. It would be recommendable to indicate the abbreviation previously in the text to improve the understanding.

We eliminated the abbreviation